

# Treatment options for Achilles tendinopathy: a scoping review of preclinical studies

Nathanael Opoku Agyeman-Prempeh[1,2,3], Huub Maas[2,4], George L. Burchell[4], Neal L. Millar[5,6], Maarten H. Moen[7,8] and Theodoor Henri Smit[1,2,3,4]

[1] University of Amsterdam, Amsterdam, Netherlands
[2] Amsterdam Movement Sciences, Amsterdam, Netherlands
[3] Department Orthopedic Surgery and Sports Medicine, Amsterdam University Medical Centre, Amsterdam, Netherlands
[4] VU University Amsterdam, Amsterdam, Noord-Holland, Netherlands
[5] University of Glasgow, Glasgow, United Kingdom
[6] Institute of Infection, Immunity and Inflammation, University of Glasgow, Glasgow, United Kingdom
[7] Department of Sports Medicine, Bergman Clinics, Naarden, the Netherlands, Unaffliated, Naarden, Netherlands
[8] High-Performance Team, Dutch National Olympic Committee & National Sports Federation, Arnhem, Netherlands

## ABSTRACT

**Background**. Achilles tendinopathy (AT) management can be difficult, given the paucity of effective treatment options and the degenerative nature of the condition. Innovative therapies for Achilles tendinopathy are therefore direly needed. New therapeutic developments predominantly begin with preclinical animal and in vitro studies to understand the effects at the molecular level and to evaluate toxicity. Despite the publication of many preclinical studies, a comprehensive, quality-assessed review of the basic molecular mechanisms in Achilles tendinopathy is lacking.

**Objectives**. This scoping review aims to summarize the literature regarding *in vitro* and *in vivo* animal studies examining AT treatments and evaluate their effect on tendon properties. Also, a quality assessment of the included animal studies is done. We provide a comprehensive insight into the current state of preclinical AT treatment research which may guide preclinical researchers in future research.

**Eligibility criteria**. Treatment options of Achilles tendinopathy in chemically or mechanically induced in vivo or in vitro Achilles tendinopathy models, reporting biomechanical, histological, and/or biochemical outcomes were included.

**Sources of evidence**. A systematically conducted scoping review was performed in PubMed, Embase.com, Clarivate Analytics/Web of Science, and the Wiley/Cochrane Library. Studies up to May 4, 2023 were included.

**Charting Methods**. Data from the included articles were extracted and categorized inductively in tables by one reviewer. The risk-of-bias quality assessment of the included animal studies is done with Systematic Review Centre for Laboratory Animal Experimentation risk-of-bias tool.

**Results**. A total of 98 studies is included, which investigated 65 different treatment options. 80% of studies reported significant improvement in the Achilles tendon characteristics after treatment. The main results were; maximum load and stiffness improvement; fibre structure recovered and less inflammation was observed; collagen

Corresponding author
Nathanael Opoku Agyeman-Prempeh, n.agyeman-prempeh@amsterdamumc.nl

I fibrils increased, collagen III fibrils decreased, and fewer inflammatory cells were observed after treatment. However, 65.4% to 92.5% of the studies had an uncertain to high risk of bias according to the risk-of-bias tool of the Systematic Review Centre for Laboratory Animal Experimentation.

**Conclusions**. Despite promising preclinical treatment outcomes, translation to clinical practice lags behind. This may be due to the poor face validity of animal models, heterogeneity in Achilles tendinopathy induction, and low quality of the included studies. Preclinical treatments that improved the biomechanical, histological, and biochemical tendon properties may be interesting for clinical trial investigation. Future efforts should focus on developing standardized preclinical Achilles tendinopathy models, improving reporting standards to minimize risk of bias, and facilitating translation to clinical practice.

## INTRODUCTION

Achilles tendinopathy (AT) is a condition that has a prevalence of approximately 6% in the general population. The condition can be induced by exercises that involve the Achilles tendon. Also, overweight people and people who are not active risk to develop AT by walking long distances or climbing the stairs often (*Weiss, 2012*). Approximately 9% of recreational runners suffer from AT which causes up to 5% of professional athletes to end their careers (*Li & Hua, 2016*; *Silbernagel, Hanlon & Sprague, 2020*). General symptoms of AT include swelling, pain, and stiffness of the posterior foot region, which may affect the quality of life, movements, and sports performances. The management of AT is challenging as many treatments lack evidence-based research and there is no gold standard (*van der Vlist et al., 2021*).

The aetiology of AT remains unclear but is associated with internal and external causes (*Tarantino et al., 2023*). Internal risk factors include biological age, tendon flexibility, prior injuries, metabolic conditions like diabetes and obesity, and genetic predispositions leading to anatomic deformities, variations in tendon morphology, and polymorphisms associated with foot injuries. External risk factors are sports practice, over-use of the Achilles tendon, hyperthermia, nutrition, medication such as corticosteroids and quinolone antibiotics, intoxication, impinging shoes or rough surfaces (*Maffulli, Sharma & Luscombe, 2004*; *Knapik & Pope, 2020*). Also, recent literature shows that the influence of psychosocial factors on the symptoms of AT should also be considered (*Edgar et al., 2022*).

As soon as tendon damage occurs, healing of the Achilles tendon is initiated in an attempt to return the tendon to homeostasis. The natural healing of the tendon takes place in three stages: the inflammatory stage, the proliferative phase, and the remodelling phase (*Li & Hua, 2016*). During these stages, tenocytes show differential expressions of among others collagen type I, II, and III, matrix metalloproteinase (MMPs), vascular endothelial growth factor (VEGFs), transforming growth factor (TGFs), and tissue inhibitors of

metalloproteinase (TIMPs) (*Li & Hua, 2016*; *Millar, Murrell & McInnes, 2017*). MMPs are proteolytic enzymes that are capable of degrading the matrix molecules. The functioning of MMP is inhibited by TIMP. VEGF regulate blood vessel formation in tendon healing. TGF is known to regulate cellular proliferation, collagen production and MMP (*Millar, Murrell & McInnes, 2017*). Due to these processes, signs of glycosaminoglycan accumulation, neovascularization, and ingrowth of nerve fibres can be observed (*Millar, Murrell & McInnes, 2017*). Understanding the effect of developed treatments for AT on the pathophysiology is needed to develop promising cures. Detailed descriptions of cells and components that are involved in the pathogenesis of AT are presented in Appendix A.

In vitro and *in vivo* animal studies greatly contribute to AT research as they allow for the detailed examination of toxicological, molecular, and cellular mechanisms, as well as biomechanical responses, at the start of developing new treatments (*Lui et al., 2011*). Many pre-clinical studies on AT have been conducted, but a review summarising all pre-clinical treatment options tested is lacking. Additionally, quality assessment of animal studies is rarely done. An overview provides a comprehensive insight into the current state of preclinical AT treatment research which may guide preclinical researchers in future research. It may also enable comparison of innovative treatment options with current treatments offered in clinical practice (*Diederich et al., 2022*). The results may show which treatments are already done in pre-clinical studies and may be interesting targets for future clinical research. The primary aim of this scoping review is to summarize the literature regarding *in vitro* and *in vivo* animal studies examining AT treatments and evaluate their effect on biomechanical, biochemical, and histological tendon properties. The secondary aim is to evaluate the quality of the included animal studies.

## MATERIALS & METHODS

A systematic search strategy was conducted by NAP and GLB (librarian). Literature selection, data extraction, and risk-of-bias evaluation were performed by a single reviewer (NAP). In case of uncertainty about literature inclusion, data extraction or risk of bias evaluation, the articles were reviewed by other authors of this paper (TS and HM).

This review was conducted without prior preregistration due to unawareness of this requirement for scoping reviews at inception of this study. Despite this, we have followed established scoping review guidelines to ensure methodological rigor and transparency. Future reviews of our group will include preregistration to align with best practices.

### Literature search strategy

A systematic search was performed in PubMed, Embase.com, Clarivate Analytics/Web of Science Core Collection, and the Wiley/Cochrane Library. The timeframe within the databases was from inception to the 4th of May 2023. The search included the following keywords and free text terms: (synonyms of) 'Achilles tendinopathy' combined with (synonyms of) '*in vivo*' or '*in vitro*'. A full overview of the search terms per database can be found in Appendix B.

## Inclusion and exclusion criteria

The review focusses in particular on Achilles tendinopathy as result of overuse. Pre-clinical models focusing on overuse AT will be included in the studies. Other Achilles tendon injuries such as ruptures will be excluded. Models based on Achilles tendon ruptures or surgically created ruptures to establish a tendinopathy model will be excluded because they are iatrogenic injuries and do not represent overuse injuries.

Furthermore, the following inclusion criteria were applied:

– Studies evaluating treatment options specifically for Achilles tendinopathy (tendinitis or tendinosis) in '*in vivo*' AT animal models or in '*in vitro*' AT tendon cells;
– The Achilles tendinopathy was chemically or mechanically induced;
– Reporting of biomechanical properties that include the description of tensile strength, elastic modulus and their relevance to the tendon function;
– Reporting of biomechanical properties in which key biochemical markers such as collagen content and inflammatory mediators are mentioned;
– Reporting of histological properties in which a detailed view of microscopic changes such as cell morphology and tendon fibre organizations is assessed.

The following exclusion criteria were applied:

– Clinical human studies;
– Systematic, scoping, narrative or other reviews;
– Commentaries;
– Guidelines;
– Treatment options for Achilles tendon rupture in both human and animal studies;
– Achilles tendon injuries induced by tenotomies or blunt trauma;
– Treatment options for Achilles tendinopathy caused by systemic conditions such as diabetes;
– If the article is published in a language other than English.

## Article selection

After completion of the initial search, the articles were uploaded to Rayyan.ai for the title and abstract screening (*Mourad Ouzzani, Fedorowicz & Elmagarmid, 2016*). Duplicates were removed with the automated duplicate screening of Rayyan and verified by NAP to confirm that a duplicate was rightfully deleted. First, titles were screened for words that fit the exclusion criteria. If the title was not clear enough, the abstract was examined. The remaining articles were full-text reviewed for meeting the inclusion criteria. Endnote was used for full-text screening as the manuscript was written in Microsoft Word with an Endnote extension for citing.

## Risk of bias quality assessment

The Systematic Review Centre for Laboratory Animal Experimentation (SYRCLE) risk-of-bias tool was used to evaluate the quality of animal studies (*Hooijmans et al., 2014*). The risk of bias was assessed on the grounds of ten points which evaluate selection bias, performance bias, detection bias, attrition bias, reporting bias, as well a category of other

sources of bias that are not covered by the SYRCLE domains. Only if a specific domain is clearly stated in the article, it was classified as low risk of bias. When a domain is stated imprecisely it was classified as unclear risk of bias. When a certain domain is not mentioned or specified it is classified as a high risk of bias. Specific attention was paid to potential conflicts of interest and included as the other source of bias (*McGuinness & Higgins, 2020*). The tables with the individual risk of bias assessment of the included animal studies are displayed in Appendix C.

### Data extraction and synthesis of results

The following data were extracted: Author and year of publication, study design, number of animals or cells, Achilles tendinopathy induction method, treatment conducted, positive and negative biomechanical, histological, and/or biochemical outcomes of the Achilles tendon before and after treatment intervention. Microsoft Word was used to create tables for data extraction. These data were labelled in the first row and the studies in the first column. During the analysis of the text, the data was extracted. After data extraction the article was analysed again to ensure data were not missed. When data was missing or an outcome measure was not reported, it was labelled as 'Not reported' (NR), and it was assumed that the authors did not evaluate a certain outcome measure.

The treatment types extracted were categorized as non-invasive, minimally invasive, invasive, or orally administered. Non-invasive treatments encompassed therapies not involving surgery or injections, but other than orally administered treatments. Examples are topical applications or laser therapies. Minimally invasive therapies are defined as treatments requiring application through injections. Oral therapies were administered orally such as diclofenac or green tea, while invasive therapies necessitated an incision or surgical procedure for administration.

The SYRCLE risk-of-bias assessment is visualized with the use of the risk-of-bias visualization tool. A high risk of bias is pictured with a red circle, uncertain risk of bias with a yellow circle and a low risk of bias with a green circle.

## RESULTS

To preserve the clarity and readability of the manuscript, which contains a large sample of heterogeneous studies, we have chosen to include articles in the main results that reported outcomes in all the domains (histological, biochemical, and biomechanical), and interventions that demonstrated significant changes. Detailed results which include the outcomes of all the included studies and the data extraction as mentioned above are provided in Appendix D, Appendix E and Appendix F. The overall outcomes and conclusions are based on the detailed results in the appendices.

### Study inclusion

The literature search yielded 4,790 results after removing duplicate articles. After title and abstract screening, a total of 335 articles were included for full-text screening. The full-text screening resulted in the inclusion of 98 articles. The selection process and exclusion reasons during the abstract and full-text review are displayed in the PRISMA flow-chart Fig. 1.

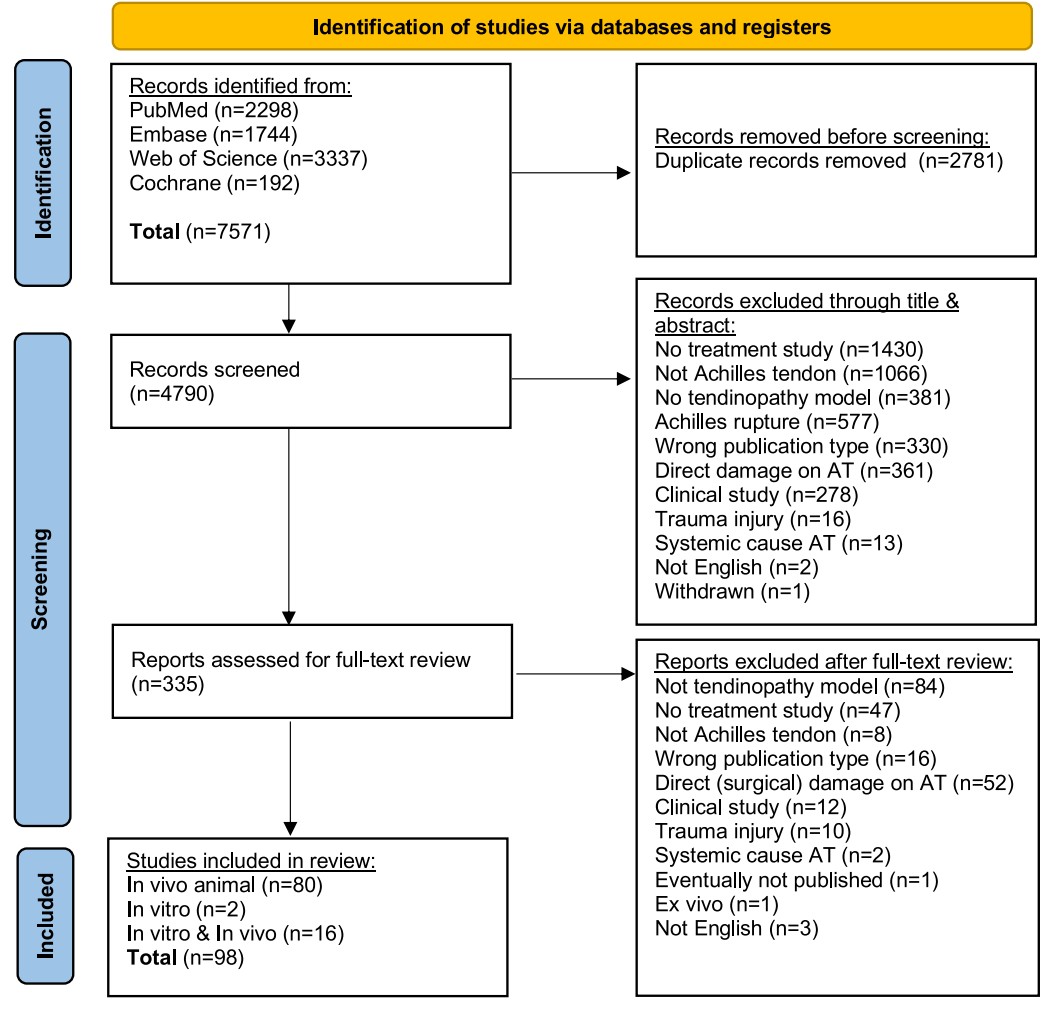

**Figure 1 Flow chart of the article selection process.**

## Description of the included studies

Detailed tables with the characteristics of the studies are presented in Appendix D, Appendix E and Appendix F. Of the included studies, 80 were *in vivo* animal studies, 16 studies involved both *in vivo* animal and *in vitro* tendon specimens, and two *in vitro* studies. In general, the studies compared the efficacy of different treatments with each other and/or with a sham group that received saline (NaCl). After saline injection, the sham and control groups of the studies showed slightly disorganized tendon fibres which turned to normal at the time of analysis. The time of analysis and follow-up time varied between the studies. Short and long term outcomes were reported whereas the shortest analysis was done after two hours and the longest follow-up time was 24 weeks. On average 48 animals were included per study with a range of 6 to 493 animals. Mostly rats were used for the analysis. After that rabbits, mice, sheep, and horses were used. The cells tested in the *in vitro* models were tenocytes derived from humans (*Lee et al., 2021*), mice (*Liu et al., 2020*),

rabbits (*Ruan et al., 2021*) sheep (*Al-Shudiefat et al., 2022*), or rat (*Lee et al., 2021*; *Choi et al., 2020a*; *Choi et al., 2020b*; *Wang et al., 2020*; *Zhao et al., 2019*; *Jeong et al., 2018*; *Kim et al., 2018*; *Chen et al., 2014*; *Vieira et al., 2018*; *Chen et al., 2012*) Achilles tendon cells.

A total of 65 different treatment interventions were evaluated (Fig. 2). The treatment types were either non-invasive, minimally invasive (intratendinous injections), invasive, or orally administered. The treatments investigated most frequently are platelet-rich plasma (PRP) ($n = 14$), low-level laser therapy (LLLT) ($n = 10$), and the administration of Non-Steroidal Anti-Inflammatory Drug (NSAID) ($n = 8$).

## Methodological quality of included studies

A total of 96 *in vivo* animal studies were analysed with the SYRCLE risk of bias tool (*Hooijmans et al., 2014*). The pooled quality of all the included studies is summarized in Fig. 3. All studies had a moderate to high risk of selection bias based on the risk-of-bias criteria. None of the studies reported the exact method of the applied randomization. There was an uncertain to high risk of performance bias. A total of 28 studies did not report how the animals were housed. The other 37 mentioned that the animals were housed but did not specify how. Only nine studies specified that the researchers giving the intervention were blinded. The risk was moderate in general regarding the detection bias. A total of 52 studies reported that random samples were used for outcome assessment and 43 studies reported that researchers analysing the outcomes were blinded. In sum, these results indicate a considerable risk of bias in the majority of the articles. The individual quality assessment of the studies is presented in Appendix C.

## Induction of tendinopathy model

The *in vivo* animal model studies describe seven methods of induction of Achilles tendinopathy. Injection with collagenase type I directly in the Achilles tendon has been done by 82 of the 97 *in vivo* animal Achilles tendinopathy models. Other studies induced AT with prostaglandin E, TGF-β1, carrageenan, PGE$_1$, betamethasone, or H$_2$O$_2$. Two studies induced the tendinopathy mechanically through intensive treadmill running for 8 or 24 weeks (*Zhang et al., 2020*; *Ng & Chung, 2012*).

Six different methods were used to establish an *in vitro* tendinopathy model. Five by adding TNF-α, IL-1β, H$_2$O$_2$, HMGB1, or bacterial lipopolysaccharides. One model used a mechanical method, by cyclic stretching of the tenocytes (*Chen et al., 2012*). Thus, while several methods exist to induce AT, induction with collagenase type I is the most common in the *in vivo* model. These induction methods led to disrupted collagen fibres, neovascularization, and infiltration of inflammatory cells.

## Reported outcomes

Detailed tables of outcomes of the included studies are presented in Appendix D, Appendix E and Appendix F. An improvement of Achilles tendon properties after treatment is reported by 80 of 98 articles. Improvement of AT such as better organization of collagen fibres, a lower amount of inflammatory cells and an improved maximum load of the tendon were mentioned. Worsening of AT after treatment was reported by seven studies. These studies evaluated the following treatments: Percutaneous augmented soft tissue

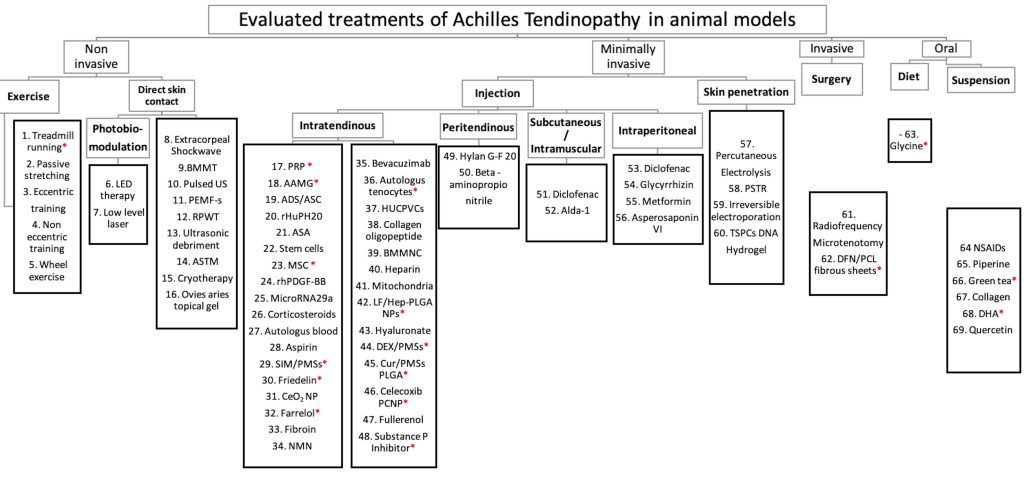

**Figure 2  Distribution of preclinical treatments.** Distribution of preclinical treatments –This figure presents the distribution of the 65 different treatment types analysed in the included studies. Abbreviations and (reference of studies evaluating specific treatment): 1. (*Bell et al., 2013*) (*Gundogdu et al., 2023*) 2. (*Ng & Chung, 2012*) 3. (*Fedato et al., 2019*) 4. (*Johnson et al., 2023*) 5. (*Godbout, Ang & Frenette, 2006*) 6. Low emitting diode (*Evangelista et al., 2021*; *Xavier et al., 2014*) 7. (*Ng & Chung, 2012*; *Marcos et al., 2011*; *Pires et al., 2011*; *Naterstad et al., 2018*; *Guerra et al., 2017*; *Marques et al., 2016*; *Torres-Silva et al., 2015*; *Marcos et al., 2014*; *Casalechi et al., 2013*) 8. (*Çinar et al., 2013*; *Chen et al., 2004*; *Tsai et al., 2013*) 9. Bone Marrow Myeloid tissue (*Naseri et al., 2008*) 10. Pulsed Ultrasound (*Martins et al., 2011*) 11. Pulsed electromagnetic fields (*Perucca Orfei et al., 2020*) 12. Radial pressure wave therapy (*Facon-Poroszewska, Kielbowicz & Przadka, 2019*) 13. (*Kamineni, Butterfield & Sinai, 2015*) 14. Augmented soft tissue mobilization therapy (*Imai et al., 2015*; *Gehlsen, Ganion & Helfst, 1999*; *Davidson et al., 1997*) 15. (*Zhang, Pan & Wang, 2014*) 16. (*Martins et al., 2011*) 17. Platelet Rich Plasma (*Ruan et al., 2021*; *Chen et al., 2014*; *Jiang et al., 2020*; *Solchaga et al., 2014*; *Dallaudière et al., 2013b*; *Li et al., 2020*; *Fedato et al., 2019*; *Facon-Poroszewska, Kiełbowicz & Prządka, 2019*; *Yan et al., 2017*; *Dallaudiere et al., 2015*; *González et al., 2016*; *Calandruccio et al., 2015*; *Dallaudiere et al., 2014*) 18. Adipose micro-grafts (*Palumbo Piccionello et al., 2021*) 19. Adipose-derived (stem) cells (*Oshita et al., 2016*; *Kokubu et al., 2020*; *Chen et al., 2018*) 20. Recombinant human Hyaluronidase (*Rezvani et al., 2020*) 21. Amniotic Suspension Allograft (*de Girolamo et al., 2019*) 22. (*Chen et al., 2014*; *Fedato et al., 2019*) 23. Mesenchymal Stromal Cells (*Lacitignola et al., 2014*; *Ahrberg et al., 2018*; *Machova Urdzikova et al., 2014*; *Crovace et al., 2008*) 24. Recombinant human platelet-derived growth factor BB (*Solchaga et al., 2014*; *Chen et al., 2018*; *Shah et al., 2013*) 25. (*Watts et al., 2017*) 26. (*Ruan et al., 2021*; *Solchaga et al., 2014*; *Naterstad et al., 2018*; *Calandruccio et al., 2015*) 27. (*Calandruccio et al., 2015*) 28. (*Wang et al., 2020*) 29. Simvastatin-loaded porous PLGA microspheres (*Jeong et al., 2018*) 30. (*Jiang et al., 2022*) 31. Cerium oxide nanoparticles (*Xu et al., 2023*) 32. (*Wu et al., 2022*) 33. (*Micheli et al., 2022*) 34. Nicotinamide mononucleotide (*Yamaura et al., 2022*) 35. (*Dallaudière et al., 2013a*; *Dallaudiere et al., 2014*) 36. (*Chen et al., 2011*) 37. Human Umbilical Cord Perivascular Cells (*Emrani & Davies, 2011*) 38. (*Ueda et al., 2008*) 39. Bone marrow mononuclear cells (*Crovace et al., 2008*) 40. (*Tatari et al., 2001*; *Williams et al., 1986*) 41. Mitochondrial Transplantation Veld (*Lee et al., 2021*) 42. Anti-inflammatory, lactoferrin-immobilized, heparin-polymeric nanoparticles (*Choi et al., 2020a*) 43. (*Yamamoto et al., 2002*) 44. Dexamethasone-containing porous microspheres (*Choi et al., 2020b*) 45. Curcumin-loaded porous PLGA (poly (D,L-lactase-co-glycoside)) microspheres (*Kim et al., 2018*) 46. Celecoxib nanoparticles (*Kim et al., 2022*) 47. (*Jiao et al., 2023*) 48. (*Ko et al., 2022*) 49. (*Tatari et al., 2004*) 50. (*Yamamoto et al., 2002*) 51. (*Marcos et al., 2011*; *Marcos et al., 2012*) 52. (*Liu et al., 2020*) 53. (*Marcos et al., 2011*) 54. (*Zhao et al., 2019*) 55. (*Zhang et al., 2020*) 56. (*Wang, Cheng & He, 2022*) 57. (*Sánchez-Sánchez et al., 2020*) 58. Percutaneous soft tissue release (*Hsieh et al., 2019*) 59. (*Wang et al., 2023*) 60. Tendon stem progenitor cells encapsulated in DNA hydrogel (*Ge et al., 2023*) 61. (*Gunes et al., 2014*) (continued on next page…)

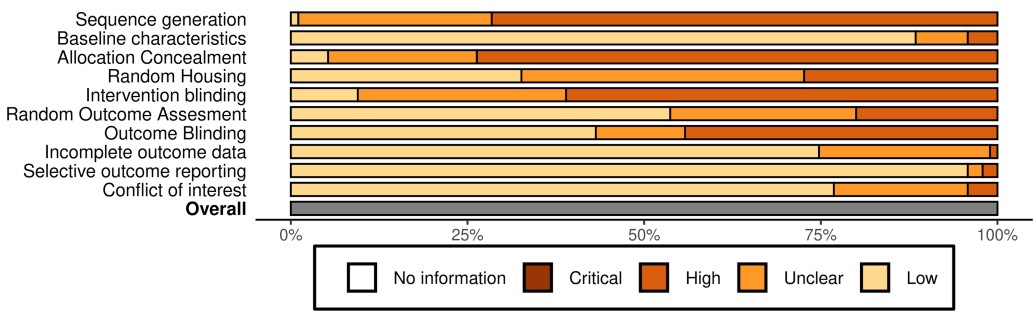

**Figure 2 (…continued)**
62. Diclofenac-immobilized polycaprolactone fibrous sheets (*Lee et al., 2019*) 63. (*Vieira et al., 2015b*; *Vieira et al., 2015a*; *Vieira et al., 2016*) 64. Non-steroidal anti-inflammatory drugs: Ibuprofen (*Bittermann et al., 2018*) Diclofenac (*Marsolais, Côté & Frenette, 2003*; *Naterstad et al., 2018*) 65. (*Gong et al., 2018*) 66. (*Vieira et al., 2015b*; (*Vieira et al., 2016*)) 67. (*Gundogdu et al., 2021*) 68. Docosahexaenoic (*Gundogdu et al., 2021*) 69. (*Semis et al., 2022*). "*" = Positive outcomes on all biochemical, histological and biomechanical outcome measures.

**Figure 3  Pooled risk of bias assessment of the included *in vivo* animal studies.**

mobilization (ASTM) (*Imai et al., 2015*), extracorporeal shockwave therapy (*Çınar et al., 2013*), diclofenac (*Marcos et al., 2011*; *Marsolais, Côté & Frenette, 2003*), passive stretching combined with laser therapy (*Ng & Chung, 2012*), early exercise after injury (*Godbout, Ang & Frenette, 2006*) and heparin (*Tatari et al., 2001*). Five studies reported that the evaluated treatment did not have effect on AT. Six studies reported mixed results (Fig. 4). Mixed results imply that an intervention positively affected a certain outcome measure, but did not affect or negatively affected another outcome measure.

Furthermore, 33 of the included studies reported the effect of their intervention on all three outcome modalities (biomechanical, histological, and biochemical). The outcomes of these 33 studies are presented in Table 1. Eighteen of these studies reported positive outcomes with their interventions in all three outcome measures and are marked with an "*" in Fig. 2 (*Choi et al., 2020a*; *Choi et al., 2020b*; *Jeong et al., 2018*; *Kim et al., 2018*; *Kim et al., 2022*; *Gundogdu et al., 2019*; *Ge et al., 2023*; *Ma et al., 2023*; *Chen et al., 2011*; *Jiang et al., 2022*; *Vieira et al., 2015b*; *Lee et al., 2019*; *Ko et al., 2022*; *Gundogdu et al., 2023*; *Wu et al., 2022*; *Oshita et al., 2016*; *Lacitignola et al., 2014*; *Jiang et al., 2020*). Other studies evaluated only one or two of these outcomes (Fig. 5). In summary, only 7% reported negative treatment effects, 12% reported mixed results, while 81% of the studies reported positive results.

### Biomechanical outcomes

The biomechanical outcomes of the treatment interventions that reported significant changes are summarized in Table 2. Generally, biomechanical properties were evaluated using rupture force, maximum load, stiffness, tensile stress, and Young's modulus. Overall, after injecting collagenase type I lower stiffness and maximum loads were reported. Several studies reported increased rupture force after intervention with 3J LED therapy

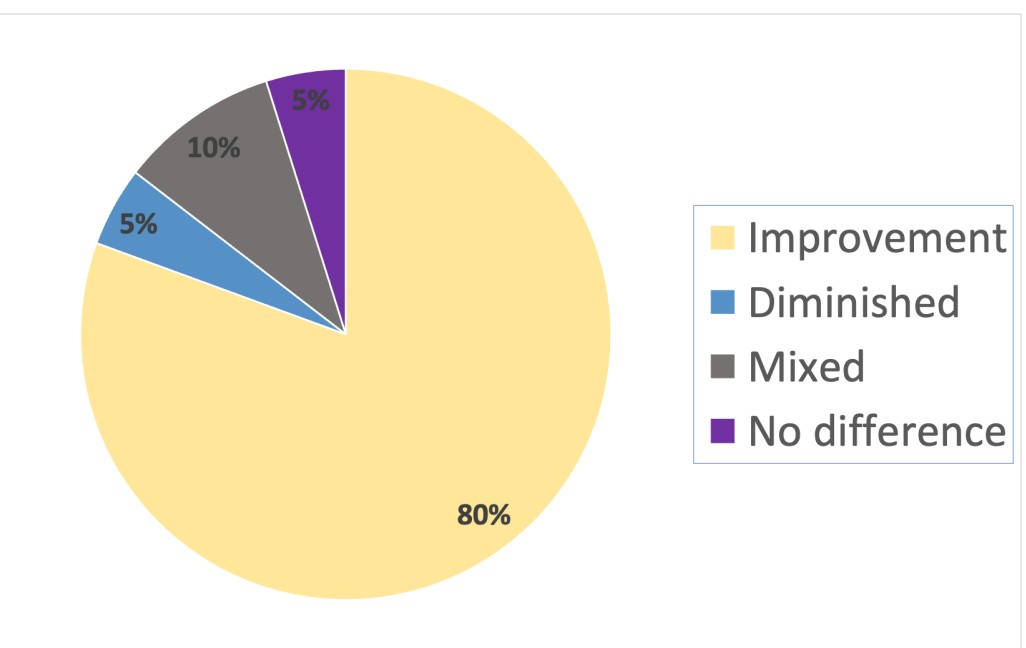

**Figure 4  Pie diagram displaying the effect of evaluated therapies on Achilles tendinopathy of the included studies.**

(*Marcos et al., 2012*), PRP (*Chen et al., 2014*), Triamcinolone combined with PRP (*Ruan et al., 2021*) glycine diet (*Vieira et al., 2015a*), green tea administration (*Vieira et al., 2015b*), recombinant human platelet-derived growth factor-BB (*Solchaga et al., 2014*), and treadmill exercise (*Bell et al., 2013*) compared to control groups. Three studies reported increased tensile modulus after intervention with treadmill exercise (*Bell et al., 2013*), HUCPVCs (*Emrani & Davies, 2011*), and rhPDGF-BB (*Solchaga et al., 2014*) compared to control and sham groups. Interestingly, early exercise after AT seems to worsen the biomechanical properties while late exercise seems to improve the biomechanical properties (*Godbout, Ang & Frenette, 2006*). The biomechanical characteristics of the Achilles tendon are significantly improved by the majority of treatments such as PRP and Low level laser therapy. However, the results also show that several treatment interventions have no effects or in fact worsen the mechanics of the Achilles tendon with therapies such as ibuprofen administration.

### Histological outcomes

For histological analysis, samples were evaluated with haematoxylin and eosin, Masson trichrome, Alcian blue, and nuclear fast red. The alignment and structure of collagen fibres were the most frequently assessed outcomes. Typically, after injecting collagenase type I, fibre disarray, and increased neovascularization were shown (*Dallaudière et al., 2013b*). Several studies reported improvement in these parameters following treatment. Injection of PRP improved the fibre structure and led to less inflammatory cells (*Jiang et al., 2020*; *Li et al., 2020*). Nevertheless, several treatments resulted in a worse condition of the Achilles

**Table 1  Results of studies that reported the effect of treatment on all outcome measures.**

| Author, year, reference | Animal used Follow-up time | Induction method | Evaluated Treatment | Histological outcome | Biochemical outcome | Biomechanical outcome |
|---|---|---|---|---|---|---|
| *Ma et al. (2023)* | Rabbits (No amount reported) 48 days | Collagenase type I injection 2400 U/2 mL | Injections with hHF-MSCs | Better ordered collagen fibres, Less inflammatory cells | Higher expression of collagen I and III, Higher expression of Tenascin-C. lower expression of MMP-9 | Upregulated maximum load |
| *Wang et al. (2023)* | 113 Rats 77 days | Collagenase type I injection 12 mg/mL 25 μL | Irreversible Electroporation | Higher number of fibroblasts and microvascular lumens, Higher cell proliferation, More spindle shaped tenocytes parallel to the collagen fibres, Less disorganized collagen fibres | Higher expression of caspase-3, $PGE_2$, TNMD-positive cells Higher proliferative activity Lower CD31, CGRP expression | Higher maximum load and tensile load Higher recovery of maximum load Lower stiffness |
| *Ge et al. (2023)* | 110 Rats 8 weeks | Collagenase type I injection 25 μL | Injection with Tendon stem progenitor cells DNA Hydrogel | Better collagen alignment. Normal level of round shaped nuclei cells Higher modified Stoll scores | Significantly higher expression of collagen type I and tenomodulin Decreased expression of collagen type III | Higher load to failure, elastic modulus and stiffness. |
| *Kim et al. (2022)* | Rats (No amount reported) 4 weeks | Collagenase type I injection 50 μL | Injection with injectable celecoxib nanoparticle hydrogels | Higher expression of hydroxyproline More enhanced collagen regeneration | Higher expression of IL-4, IL-10 Lower expression of COX-2, IL-1, IL-6, MMP-3, MMP-13, and TNF-$\alpha$ | Higher stiffness value and tensile strength |
| *Jiang et al. (2022)* | Rats (No amount reported) 4 weeks | Collagenase type I injection 20 μL | Injection with Friedelin | Increased structural order of tendons, reduced inflammatory cells, better alignment collagen fibres, reduces neo-vascularization, high-strength produced collagen fibres | Decreased expression of Dcn, Scx, Mkx, Tnmd, F4/80+, Il-6, TNFa and IL1-B | Increased failure load and ultimate stress |
| *Xu et al. (2022)* | 40 Rats 4 weeks | Collagenase type I injection 25 μL | Injection with exosomes and ectosomes isolated from Adipose-derived mesenchymal stem cells | Better histological score Less inflammation and spindle like cells Tighter fibre structure and less angiogenesis | Decreased expression of collagen type 3 | Better failure load and ultimate tensile strength |
| *Wu et al. (2022)* | 24 Rats 4 weeks | Collagenase type I injection 30 μL | Injection with Farrerol | Less low stretch stress fibers | Higher expression of tnmd, scx and mkx Lower mRNA levels of the pro-inflammatory cytokines Mcp1, Pge2, Tnfa, Il-1b, Il-6 and Il-17 | Higher Young's modulus and maximum stress |
| *Ko et al. (2022)* | 28 Rats 5 weeks | Collagenase type I injection 20 μL | Injection with Substance P inhibitor | Lower collagen disruption Lower proteoglycans and glycosaminoglycan's deposition | Decreased expression of IL-6 and NK1R | Higher tensile strength |
| *Gundogdu et al. (2021)* | 40 Rats 8 weeks | Collagenase type I injection 500 UI | Oral gavaged with Docosahexaenoic acid | Fibroblast and fibrocytes proliferation Less degeneration | Higher expression of collagen type I | Higher ultimate tensile force, yield force and stiffness |
| *Ruan et al. (2021)* | 33 Rabbits 8 weeks | Collagenase type I injection 300 UI 260 U/mg | Injection with Leukocyte-poor PRP Or Triamcinolone acetonide in combination with Leukocyte-poor PRP 200 μL | More vascular infiltration, higher cell density, more small disordered collagen fibers Triamcinalone+PRP histological score almost same when compared to normal group | Increased CHI3L1, MMP1, and MMP12, TNFRSF1B and HMOX1 (anti-apoptotic) Upregulated S100A12, IL1A, IL1B, and IL7 | Maximum tension load greater in the treatment group |

**Table 1** (*continued*)

| Author, year, reference | Animal used Follow-up time | Induction method | Evaluated Treatment | Histological outcome | Biochemical outcome | Biomechanical outcome |
|---|---|---|---|---|---|---|
| *Palumbo Piccionello et al. (2021)* | 16 Sheep 8 weeks | Collagenase type IA injection 500 UI | Injection of Adipose-autologous micro grafts | Lower presence of necrosis, damaged fibers, and inflammatory infiltrative process. Lower aspect of edema and myxoid | Higher expression of collagen type I, FVIII (more active neo-angiogenesis) Lower expression of collagen type III, TGF-β1 | Slightly higher maximum load and rupture force |
| *Choi et al. (2020a)* | 28 Rats 4 weeks | Collagenase type I injection 50 μL | Injection with anti-inflammatory, Lactoferrin-immobilized, heparin-polymeric nanoparticles | Prevented disruption of collagen fibrils | Decreased mRNA levels of pro-inflammatory factors and proteases. | Greatly increased stiffness and tensile strength |
| *Jiang et al. (2020)* | 28 Rabbits 6 weeks | Collagenase type I injection 110 μL | Injection of leukocyte poor-PRP or leukocyte rich PRP | Lr-PRP compared to Lp-PRP and saline group: Better fibre structure and less angiogenesis in the Lr-PRP group compared to the Lp-PRP group More mature collagen fibers Lp-PRP compared to saline: Lower histological scores overall | Lr-PRP compared to Lp-PRP and saline group: Higher failure load, stiffness, and tensile stress after 6 weeks | Lr-PRP compared to Lp-PRP and saline group: Higher failure load, stiffness, and tensile stress after 6 weeks |
| *Choi et al. (2020b)* | 24 Rats 4 weeks | Collagenase type I injection 50 μL | Injection with Dexamethasone-containing porous microspheres (DEX/PMSs) | Decreased collagen fibre breakdown DEX(10%)/PMS displayed the best therapeutic effect | Decreases the level of COX-2, IL-1β, IL-6, and TNF-α | Tensile strength and stiffness increased dose-dependently in the DEX/PMSs treated groups |
| *Wang et al. (2020)* | 24 Rats 5 weeks | Collagenase type I injection 30 μL | Intratendinous injection with Aspirin | Better arrangement of collagen fibers | Higher expression of TNC, TNMD, and SCX | Better ultimate stress and young modulus |
| *Li et al. (2020)* | 32 Rabbits 6 weeks | Collagenase type I injection 300 UI/rabbit, 260 u/mg | Injection with autologous leukocyte rich-PRP | Better healing results and histology score (Better fibre arrangement, structure, angiogenesis, rounding of nuclear, inflammation, and cell density) | Higher expression of IL-6, Il-10 | Failure load greater in the control group |
| *Rezvani et al. (2020)* | 493 mice 25 days | Injection with 2 doses rHuTGF-β 100ng | Injection in retro calcaneal bursa with Recombinant Human Hyaluronidase | After 9 days reduced amount of glycosaminoglycan's Rapid and extensive clearance of accumulated aggrecan/hyaluronan | An increased expression of Ier3, Rel, Tlr2, Tnfrsf1b ,Adora2b, Cdh1, Dcn, Has1, Wisp1, Pkm , Ccl2, Ccl7, Cd80, Cxcl10, F10, Infb1, Mif, Il12a and Ptx3 | Decreased maximum loads |
| *Lee et al. (2019)* | Rabbits (No exact amount) 4 weeks | Collagenase type I injection 50 μL | Surgical placement of 1 and 5mg diclofenac-immobilized polycaprolactone (DFN/PCL) fibrous sheets (3 × 2cm) | Decreased number of inflammatory cells Restored collagen fibre arrangement | Decreased expression of inflammatory cytokines | Better stiffness and tensile strength of the tendon tissues |
| *De Girolamo et al. (2019)* | 72 Rats 4 weeks | Collagenase type I injection 3 mg/mL 185 IU/mg | Injection with Amniotic suspension Allograft | Improvement in tissue structure, fibre alignment, fibre organization, cell density, and fatty deposit formation | More abundant presence of residual human nuclei | Better maximum load values |
| *Fedato et al. (2019)* | 41 Rats 4 weeks | Collagenase type I injection 250 IU 10 mg/ml | Injection with: -Stem cells out of 2ml blood -Platelet-rich plasma out of 1-2 ml blood | No significant differences | Stem cell group highest percentage of collagen type I | No difference between groups in tensile and yield strength |

**Table 1** (*continued*)

| Author, year, reference | Animal used Follow-up time | Induction method | Evaluated Treatment | Histological outcome | Biochemical outcome | Biomechanical outcome |
|---|---|---|---|---|---|---|
| *Bittermann et al. (2018)* | 307 Mice 25 days | Injection with active rHuTGF-β1 100 ng | Oral administration of Ibuprofen | Elevated levels of chondroid Increased blood vessels in the adjacent fat pad Expression of multiple groups of GSI-positive cells Delayed clearance of pro-inflammatory matrix Prolonged ECM remodelling | Delayed time to normalization of NF □b target and wound-healing genes A much higher expression of Cxcl5, Col3a1, Il6, Mmp9, Col5a1, Cxcl3 and Ptgs2 genes | Loss in stiffness and elastic modulus |
| *Jeong et al. (2018)* | 24 Rats 7 weeks | Collagenase type I injection 50 µL | Injection with Simvastatin-loaded porous PLGA microspheres | Suppressed collagen matrix disruption | Decreased levels of MMP-3, COX 2, IL-6, and TNF-β | Better stiffness and tensile strength |
| *Kim et al. (2018)* | 28 Rats 7 weeks | Collagenase type I injection 50 µL | Injection with Curcumin-loaded porous PLGA microspheres | Prevented collagen disruption Repaired collagen matrix organization in a dose-dependently manner | Decreased expression of MMP-3, MMP-13, COX-2, ADAMTS-5, IL-6, and TNF-a | Better tensile strength |
| *Vieira et al. (2016)* | 50 Rats 22 days | Collagenase type I injection 10 µL | Oral administration of: -Green tea leaves of Camellia sinensis - Diet containing 5% Glycine | No different histological outcomes between groups | The highest concentration of glycosaminoglycan's in Green tea + Glycine group, almost similar to the control group | Better maximum load almost similar to control in green tea group |
| *Vieira et al. (2015a)* | 42 Rats 21 days | Collagenase type I injection 10 µL | Oral diet containing 5% glycine | Thicker epitenon observed | Higher amount of hydroxyproline almost similar to control Lowest concentration of non-collagenous proteins | Greater maximum load |
| *Imai et al. (2015)* | 12 Rabbits 21 days | Collagenase type I injection 30 µL | Percutaneous Augmented Soft tissue mobilization | More aligned collagen fibers | Decreased level of collagen type III fibers | Lower storage modulus No difference in loss tangent |
| *Chen et al. (2014)* | 18 Rats 6 weeks | Collagenase type I injection 250 UI | Injection with: -Allogenic PRP -Allogenic Tendon derived stem cells -Combination (PRTD) | PRTD compared to other treatment groups Lower nuclear rounding scores Better fibre structure, arrangement, and inflammation scores | PRTD compared to Tend and other treatment groups An increased expression of collagen type I, Scx, Tenascin C Decreased expression of Runx2, PPARy, and SOX9 | PRTD compared to Tendinitis Better maximum load and stiffness |
| *Machova Urdzikova et al. (2014)* | 81 Rats 5 weeks | Collagenase type I injection 250 UI 0.3 mg | Injection with Mesenchymal stromal cells | Lower cellularity and more spindle-shaped cells Decreased vascularity Better organization of collagen fibers Denser tissue matrix | Higher expression of collagen type I and III No difference in aggrecan and versican expression | No difference in mean peak force Increase in stiffness |
| *Bell et al. (2013)* | 88 Mice 4 weeks | Injection with 100ng active TGF-β1 | Uphill treadmill running at 32cm/s, for 20min/day. 5 days/week for 2 or 4 weeks | Treadmill exercise prevented groups of rounded cells, with enlarged and rounded nuclei, and with each cell surrounded by its organized pericellular matrix. | Reduction of collagen type I, collagen type II and collagen type III Reduction Can, Agg, Adamts5 and MMP-3 expression | After 4 weeks, recovery in maximum load, stiffness, maximum stress, and tensile modulus |
| *Chen et al. (2011)* | 44 Rabbits 4 weeks | Collagenase type I injection 30 µL | Injection with Autologous tenocytes therapy-A: Tenocytes harvested from patellar epitendineum tissue | Better histology scores regarding fibre structure, arrangement, rounding of nuclei and inflamed cells at 8 weeks Reduced angiogenesis | An increased expression of collagen type I No difference in collagen type III expression | Higher ultimate failure load and mean stiffness |

**Table 1** (*continued*)

| Author, year, reference | Animal used Follow-up time | Induction method | Evaluated Treatment | Histological outcome | Biochemical outcome | Biomechanical outcome |
|---|---|---|---|---|---|---|
| *Emrani & Davies (2011)* | 48 Rats 1 month | Collagenase type I injection 30 $\mu$ L | Injection with Human Umbilical Cord Perivascular cells | More linear collagen fibre arrangement at 30 days | Morphology of the cells turned elongated coming from an ovoid form | Higher tensile strength and young modulus |
| *Chen et al. (2004)* | 123 rats 12 weeks | Collagenase type I injection 30 $\mu$ L | Percutaneous Extracorporeal Shock Wave treatment | Granulation tissue and inflamed cell infiltration improved Increased vascularity and newly formed tendon tissue seen | Increased PCNA expression Increase in cell proliferation An increased expression of TGF-$\beta$1 and IGF-1 at 1 and 4 weeks | Better mechanical load to failure and stiffness. However more than 200 pulses decreased biomechanical properties |
| *Marsolais, Côté & Frenette (2003)* | Rats (No exact amount) 28 days | Collagenase type I injection 30 $\mu$ L | Oral administration of diclofenac dissolved in water | Collagen fibers remained small and disorganized compared to control | -Reduced accumulation of PMN and ED1 $^+$ at day 1 -No effect on PMN and ED1 $^+$ in the core of tendon at day 28 | Diclofenac treatment worsened biomechanical properties |

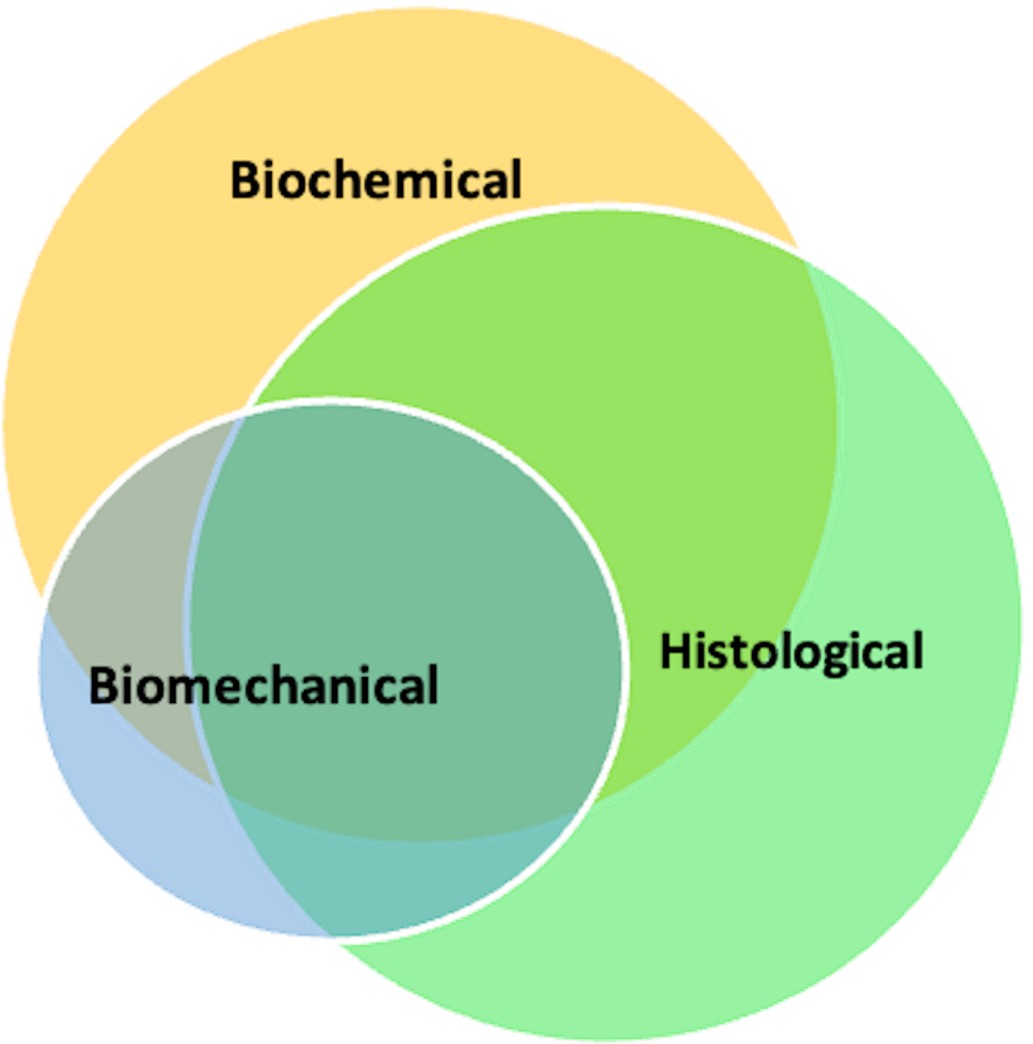

**Figure 5** **Venn diagram displaying the overlap of outcome measures of the included studies.**

 

**Table 2  Biomechanical effects on Achilles tendon tissue.**

| Biomechanical Outcomes | Rupture force | Maximum load | Stiffness | Tensile stress | Young's modulus |
|---|---|---|---|---|---|
| Docosahexaenoic (*Gundogdu et al., 2021*) | | + | + | + | |
| AAMG (*Palumbo Piccionello et al., 2021*) | + | | | | |
| PRP (*Chen et al., 2014*; *Jiang et al., 2022*; *Solchaga et al., 2014*; *Li et al., 2020*; *Yan et al., 2017*; *Calandruccio et al., 2015*) | | + | + | + | |
| rHuPH20 (*Rezvani et al., 2020*) | | ↓ | ↓ | | ↓ |
| Ibuprofen (*Bittermann et al., 2018*) | = | = | ↓ | = | ↓ |
| hADSC+ rhPDGF-BB (*Chen et al., 2018*) | | + | + | + | |
| Glycine (*Vieira et al., 2015a*) | | + | | | |
| Green tea (*Vieira et al., 2015b*) | | + | | | |
| ASTM (*Gehlsen, Ganion & Helfst, 1999*; *Davidson et al., 1997*) | | | | | ↓ |
| rhPDGF-BB (*Solchaga et al., 2014*) | | + | + | + | + |
| hMSC (*Ma et al., 2023*; *Machova Urdzikova et al., 2014*) | = | | + | | |
| LLLT (*Marcos et al., 2012*; *Naterstad et al., 2018*; *Marcos et al., 2014*) | | +/↓ | +/↓ | | |
| Uphill treadmill (*Gundogdu et al., 2023*; *Bell et al., 2013*) | + | + | + | + | + |
| Extracorporeal shockwave (*Chen et al., 2004*; *Yoo et al., 2012*) | | ↓ | +/↓ | | |
| Autologous tenocytes (*Chen et al., 2011*) | | + | + | | |
| HUCPVCs (*Emrani & Davies, 2011*) | | | | + | + |
| Early exercise (*Godbout, Ang & Frenette, 2006*) | ↓ | | | ↓ | |

| Biomechanical Outcomes | Rupture force | Maximum load | Stiffness | Tensile stress | Young's modulus |
|---|---|---|---|---|---|
| LF/Hep-PLGA NPs (*Choi et al., 2020a*) | | | + | + | |
| DEX/PMSs PLGA (*Choi et al., 2020b*) | | | + | + | |
| Aspirin (*Wang et al., 2020*) | | | | + | + |
| Diclofenac (*Marcos et al., 2011*; *Naterstad et al., 2018*) | ↓ | ↓ | | | |
| Cur/PMSs PLGA (*Kim et al., 2018*) | | | | + | |
| SIM/PMSs PLGA (*Jeong et al., 2018*) | | | + | + | |
| Irreversible Electroporation (*Wang et al., 2023*) | | + | ↓ | + | |
| US + Ximenia Americana L (*Leal et al., 2016*) | + | | | | |

**Notes.**

+, Improvement of biomechanical properties; ↓, Deterioration of biomechanical properties; +/ ↓, Both improvement and deterioration reported; =, No difference reported
Blank boxes implies that the biomechanical outcome is not reported for the treatment.

tendon. *Sánchez-Sánchez et al. (2020)* reported that percutaneous electrolysis showed more signs of inflammation and collagen fibre disarray. Also, treatment interventions that had improvement in certain histological aspects and deterioration in other aspects were reported. The use of low-level laser therapy showed improvement in the collagen alignment of the tendon, however, it also led to more signs of inflammation in the cells (*Marcos et al., 2011*; *Marcos et al., 2012*; *Xavier et al., 2010*; *Pires et al., 2011*). Other reported significant histological changes in the treatment interventions are reported in Table 3. The histological properties of the Achilles tendon are significantly improved by the majority of treatments. However, several treatments negatively affected cells or showed both worsening and improvements depending on the histological characteristics that were evaluated.

**Biochemical outcomes**

Biochemical analyses were performed by immunohistochemistry, gene expression analysis (PCR), and RNA sequencing (Table 4). Approximately 29% of the studies reported the influence of MMP expression. Some studies reported that PRP, Triamcinolone, oral Ibuprofen, and Glycine increase the expression of MMP in comparison with control. Treatment with low-level laser therapy suggested mixed results on the expression of MMP. Several growth factors were assessed by the included studies. Nine studies reported the effect of VEGF and seven studies the effect of the TGF. PRP, LLLT, and Percutaneous electrolysis were treatments that had a bigger effect on VEGF when compared with the control group.

About 15% of the included articles reported effects of treatment on interleukins (IL). The intervention of several studies that evaluated LLLT, Simvastatin, PRP, and tumor necrosis factor-alpha (TNF-β) resulted in a reduced turnover of IL-6, and IL-1β, and

**Table 3** Histological effects on Achilles tendon tissue.

| Histological outcomes | Bonar & movin scores | Fiber alignment | Vascularity | Inflammatory cell infiltration | Fibroblast count | Fat cells infiltration |
|---|---|---|---|---|---|---|
| LED (*Evangelista et al., 2021*) | + | + | + | | + | + |
| PEMF-s (*Perucca Orfei et al., 2020*) | + | + | | + | | |
| RPWT + ADSC + PRP (*Facon-Poroszewska, Kiełbowicz & Prz̨adka, 2019*) | + | + | + | | + | |
| Docosahexaenoic (*Gundogdu et al., 2021*) | + | + | + | ↓ | + | |
| Ultrasonic debridement (*Kamineni, Butterfield & Sinai, 2015*) | + | + | | | | |
| PRP (*Chen et al., 2014*; *Solchaga et al., 2014*; *Li et al., 2020*; *Yan et al., 2017*) | + | + | | + | | |
| ADS/ASC (*Oshita et al., 2016*; *Kokubu et al., 2020*) | + | + | | | + | |
| rHuPH20 (*Rezvani et al., 2020*) | + | | | | | |
| ASA (*De Girolamo et al., 2019*) | + | + | | | | + |
| MSC (*Machova Urdzikova et al., 2014*) | + | | | ↓ | | |
| MicroRNA29a (*Watts et al., 2017*) | + | + | | | | |
| Autologous tenocytes (*Chen et al., 2011*) | + | + | | | | |
| Extracorporeal shockwave (*Chen et al., 2004*; *Yoo et al., 2012*) | | + | + | + | | |
| Percutaneous electrolysis (*Sánchez-Sánchez et al., 2020*) | | + | + | ↓ | | |
| Radiofrequency microtenotomy (*Gunes et al., 2014*) | | | + | | | |
| Ibuprofen (*Bittermann et al., 2018*) | | | + | + | | |
| BMMT (*Naseri et al., 2008*) | | + | | + | | |
| rhPDGF-BB (*Solchaga et al., 2014*) | | + | | + | | |
| AAMG (*Palumbo Piccionello et al., 2021*) | | | | + | | |
| Cur/PMSs PLGA (*Kim et al., 2018*) | | + | | + | | |

| Histological outcomes | Bonar & movin scores | Fiber alignment | Vascularity | Inflammatory cell infiltration | Fibroblast count | Fat cells infiltration |
|---|---|---|---|---|---|---|
| Hylan G-F 20 (*Tatari et al., 2004*) | | + | | + | | |
| PSTR (*Hsieh et al., 2019*) | | | | + | | |
| Early Exercise (*Godbout, Ang & Frenette, 2006*) | | | | ↓ | | |
| LLLT (*Marcos et al., 2012*; *Naterstad et al., 2018*; *Marcos et al., 2014*) | | + | | ↓ | | |
| Diclofenac (*Marcos et al., 2011*; *Naterstad et al., 2018*) | | | | ↓ | | |
| Heparin (*Tatari et al., 2001*) | | +/↓ | | ↓ | | |
| ASTM (*Gehlsen, Ganion & Helfst, 1999*; *Davidson et al., 1997*) | | | | | + | |

**Notes.**

+, Improvement of histopathology; ↓, deterioration of histopathology; +/ ↓, Both improvement and deterioration reported

Blank boxes implies that the histological outcome is not reported for the treatment.

increased turnover of IL-4, IL-13, and IL-10. The effect of the treatment on cyclooxygenase (COX) and tumour necrosis factor (TNF) expression is reported by 26 studies. Nine studies reported a decreased expression of both COX-2 and TNF-β. In sum, many biochemical cells and proteins were evaluated with mixed effects. The cells and proteins evaluated differ greatly per study. Most of the biochemical effects were evaluated by studies that treated the Achilles tendinopathy model with either LLLT or PRP.

## DISCUSSION

Achilles tendinopathy is a challenging condition to treat as many treatment options lack scientific evidence and have diverse results in clinical practice (*van der Vlist et al., 2021*). Many pre-clinical studies concerning the treatment of AT have been done to contribute to this challenging field. This scoping review aims to summarize the literature regarding *in vitro* and *in vivo* animal studies of AT treatments and to evaluate the quality of these studies. A total of 98 studies were included which analysed 65 different treatments of which 80% reported promising results regarding the biomechanical, histological, and biochemical outcomes. However, the included studies had a moderate to high risk of bias. The variety of available data and the quality of the studies display the challenging preclinical research domain for AT treatments.

### Preclinical models of Achilles tendinopathy

Currently, there is no animal model that accurately mimics AT in humans (*Perucca Orfei et al., 2016*; *Warden, 2007*). Considering the disadvantages and advantages of the different animal models, the sheep and rabbit models appear to be the better overall option.

**Table 4  Biochemical effects on collagenous and non-collagenous proteins of the Achilles tendon.**

| Biochemical outcomes | Col type I | Col type II | Col type III | Col type IV | Col type X | MMP 1 | MMP 2 | MMP 9 | MMP 13 | VEGF | TGF-β | TIMP | ADAMTS | Dcn | Agg | Tnc |
|---|---|---|---|---|---|---|---|---|---|---|---|---|---|---|---|---|
| LLLT (*Marcos et al., 2012*; *Pires et al., 2011*; *Guerra et al., 2017*; *Marques et al., 2016*; *Marcos et al., 2014*; *Casalechi et al., 2013*) | + | | ↓ | | | ↓ | | +/↓ | +/↓ | + | + | | | | | |
| PRP (*Chen et al., 2012*; *Jiang et al., 2020*; *Li et al., 2020*; *Yan et al., 2017*; *González et al., 2016*) | + | | + | | | +/↓ | | | | + | | + | | | | |
| PRP + TDSC (*Chen et al., 2014*) | | | | | | | | | | | | | | | | + |
| rhPDGF-BB + ADSC (*Chen et al., 2018*) | + | | | | | | | | | | | | | | | |
| ADSC (*Oshita et al., 2016*) | + | | ↓ | | | | | | | + | | | | | | |
| MSC (*Lacitignola et al., 2014*; *Ahrberg et al., 2018*; *Machova Urdzikova et al., 2014*) | + | | + | | | | | | | | | | | | | = |
| Docosahexaenoic (*Gundogdu et al., 2021*) | + | | | | | | | | ↓ | | | | | | | |
| CBMSCs (*Crovace et al., 2008*) | + | | ↓ | | | | | | | | | | | | | |
| BMMNCs (*Crovace et al., 2008*) | + | | ↓ | | | | | | | | | | | | | |
| Autologous Tenocytes (*Chen et al., 2011*) | + | | | | | | | | | | | | | | | |
| Green tea (*Vieira et al., 2015b*) | + | | | | | | | | | | | | | | | |
| Glycine (*Vieira et al., 2018*; *Vieira et al., 2015a*) | + | | | | | | +/↓ | + | | | | | | | | |
| Green tea + glycine (*Vieira et al., 2015b*; (*Vieira et al., 2016*)) | + | | | | | | + | +/↓ | | | | | | | | |
| Treadmill Exercise (*Bell et al., 2013*) | ↓ | + | ↓ | | | | | | | | | + | | | + | |
| Metformin (*Zhang et al., 2020*) | | ↓ | | | | | | | | | | | | | | |
| Ultrasonic Debridement (*Kamineni, Butterfield & Sinai, 2015*) | | | | | = | | | | | | | | | | | |
| Radiofrequency Microtenotomy (*Gunes et al., 2014*) | | | | = | | | | | | | | | | | | |
| ASTM (*Imai et al., 2015*) | | | | ↓ | | | | | | | | | | | | |
| Piperine (*Gong et al., 2018*) | | | ↓ | | | | ↓ | ↓ | | | | | | | | |
| Percutaneous Electrolysis (*Sánchez-Sánchez et al., 2020*) | + | | | + | | | + | + | | + | | | | | | |
| Hyaluronate (*Wu et al., 2016*) | + | | + | | | | | | | | | | | | | |

**Table 4** (*continued*)

| Biochemical outcomes | Col type I | Col type II | Col type III | Col type IV | Col type X | MMP 1 | MMP 2 | MMP 9 | MMP 13 | VEGF | TGF-β | TIMP | ADAMTS | Dcn | Agg | Tnc |
|---|---|---|---|---|---|---|---|---|---|---|---|---|---|---|---|---|
| LED (*Xavier et al., 2014*) | + | | + | | | | | | | | | | | | | |
| Triamcinolone + PRP (*Ruan et al., 2021*) | | | | | | + | | | | | | | | | | |
| AAMG (*Palumbo Piccionello et al., 2021*) | | | | | | ↓ | | | | + | + | | | | | |
| Mitochondrial transplantation (*Lee et al., 2021*) | | | | | | ↓ | | | | | | | | | | ↓ |
| Collagen Oligopeptide (*Ueda et al., 2008*) | | | | | | | ↓ | | | | | | | | | |
| Oral Ibuprofen (*Bittermann et al., 2018*) | | | | | | | | + | | | | | | | | |
| Cur/PMSs PLGA (*Kim et al., 2018*) | | | | | | | | | ↓ | | | + | | | | |
| SIM/PMSs PLGA (*Jeong et al., 2018*) | | | | | | | | | ↓ | | | | | | | |
| Extracorporeal shockwaves (*Chen et al., 2004*) | | | | | | | | | | | + | | | | | |
| rHuPH20 (*Rezvani et al., 2020*) | | | | | | | | | | | | | | + | | |

**Notes.**

+, Increased expression in tendon; ↓, Decreased expression in tendon; +/↓, Both decreased and increased expression reported; =, No difference reported
Blank boxes imply that the biochemical outcome is not reported for the treatment; Col, Collagen; MMP, Matrix metalloproteinase; VEGF, Vascular endothelial growth factor; TGF-β, Transforming growth factor beta; ADAMTS, a disintegrin and metalloproteinase with thrombospondin motifs; Dcn, Decorin; Agg, Aggrecan; Tnc, tenascin-C.

Rabbit's models may be more comparable to humans as their cellular and tissue physiology approximates that of humans. Sheep models have weight bearing of the tendon similar to humans (*Zhang et al., 2022*). The higher representation of rats (66% of the included studies) could be explained by their easy availability and low cost. The disadvantage is the small size of their tendons which makes intratendinous injections more challenging. Due to their size histological and biomechanical analyses are also more complex (*Lui et al., 2011*). Additionally, rat tendons have a much higher surface/volume ratio, which may overestimate the effect of treatment. It is important to highlight that the majority of evaluated animal models are quadruped models which differs greatly from bipedal models. The lack of a low-cost valid animal model for AT hinders representative high-quality pre-clinical research, which can eventually be translated into clinical practice.

The variations of the induction methods in animal studies contributes to the heterogeneity of the included studies. The predominant chemical induction (86% of studies) involves intratendinous injection of collagenase type I which disrupts collagen bundles that mimic human AT. Currently, a comprehensive protocol for establishing AT with collagenase is lacking, leading to inconsistency of applies collagenase doses. This is characterized by the diverse doses applied (1 mg/mL–10 mg/mL) among rats in the included studies of this scoping review. *Perucca Orfei et al. (2016)* reports that higher doses (3 mg/mL) demonstrate a closer resemblance to human disease, displaying features like fatty deposits and morphological changes similar to human AT at day 15. Additionally, a disadvantage of induction with collagenase type I is that tendon damage is immediately apparent after induction which may not resemble the pathophysiology of overuse AT in humans (*Warden, 2007*). Mechanical overloading, a potentially more

valid induction method, is less frequently applied (two studies) possibly due to time and resource limitations (*Zhao et al., 2019*; *Ng & Chung, 2012*). To enhance the validity and comparability of preclinical AT models, there is a need for a validated animal model and standardized induction methods.

*In vitro* studies of AT are mostly done at an early phase to evaluate the toxicity, cell differentiation, pathophysiology, and different AT pathways. Interestingly, only one study was found that used human tenocytes to establish an *in vitro* AT model by administering TNF-β (*Lee et al., 2021*). However, the article did not state whether these tenocytes were derived from patients with AT or healthy Achilles tendons. It would be interesting to investigate if the use of human tenocytes gives more representative outcomes than animal tenocytes. Furthermore, the specific type of human tenocytes utilized in studies is crucial, as differences exist between energy-storing tendons and positional tendons (*Birch, 2007*). The Achilles tendon is an example of an energy storing tendon which is required to stretch and recoil to provide an adequate return of energy. While positional tendons such as the tibitalis anterior tendons are composed of more stiff matrix. The differences between energy-storing tendons such as the Achilles tendon and positional tendons such as the tibialis anterior may be interesting to study as they have different functions. However, it is essential to clearly distinguish these aspects in order to accurately assess the results of such studies.

This scoping review excluded one study that used an *ex vivo* model of freshly harvested bovine superficial digital flexor tendons to evaluate the use of genipin injections for collagenase D-induced AT (*Tondelli et al., 2020*). The use of *ex vivo* models is interesting as animals have multifunctional use and potentially do not have to be bred specifically for AT research. As ethical and moral subjects about the necessity and efficiency of the use of animal models are ongoing, the use of validated *ex vivo* AT models could be an answer to this issue.

## Evaluated therapies in animal models

Interestingly, there is heterogeneity in the most assessed treatments (PRP, LLLT, and NSAID) within themselves. For example, the method of preparing PRP and its composition differs per study. *Jiang et al. (2020)* and *Li et al. (2020)* compared the use of leukocyte-rich PRP and leukocyte-poor PRP with each other. Both studies found that the use of leukocyte-rich PRP achieved better results than leukocyte-poor PRP at the early stage of AT. However, others found that late-stage leukocyte-poor PRP is more beneficial for AT (*Yan et al., 2017*). Furthermore, photo-bio modulation therapies are delivered in different wavelength intensities and frequencies in the included studies. Additionally, pulsed-based therapies are delivered in different wave- and radio frequencies. There are also studies reporting conflicting results with intratendinous injection with heparin. Intratendinous injections with heparin were reported to result in more pronounced collagen fibers, less cellularity, and less neovascularization (*Williams et al., 1986*). However, others found that heparin had a degenerative effect on the Achilles tendon (*Tatari et al., 2001*). All in all, even though the same treatment types are used, outcomes may vary because the specifics of the intervention differ.

Interestingly, treatment methods for AT which are contraindicated in clinical practice because of their adverse events such as glucocorticoids and NSAID, were used in innovative administration tools. *Choi et al. (2020b)* used porous microspheres to administer dexamethasone, which resulted in healing and anti-inflammatory effects and resulted in no sign of degeneration. *Lee et al. (2019)* implanted a diclofenac-immobilized polycaprolactone fibrous sheet through surgery. This resulted in higher collagen content, anti-inflammatory effects, and improved mechanical strength. These innovative options may create opportunities for methods that were not applicable because of their initially adverse effects, to be reconsidered and expand the treatment options.

## Translation and comparability of preclinical studies

In 2018 and 2021 the AT management guidelines of The Orthopaedic Section of the American Physical Therapy Association (APTA) and Dutch multidisciplinary guidelines on Achilles tendinopathy were published (*Martin et al., 2018*; *de Vos et al., 2021*). There is not much overlap when comparing these guidelines with the preclinical treatment options included in this scoping review. Both guidelines advocate the use of exercise therapies to manage AT. However, only about 4% of the included treatments in this review evaluated exercise therapies for AT. That said, the lack of exercise based animal studies could be attributed to the challenges of implementing exercise regimens in animal models. These numbers indicate that the translation of preclinical to clinical concepts lacks synchronization. More consideration should be drawn to improving the translatability of these studies. Embracing open science initiatives where clear guidelines for pre-clinical research are developed may contribute to the translatability of this research field (*Diederich et al., 2022*).

Preclinical treatments that show positive effects on the biomechanical, histological, and biochemical outcome measures warrant consideration for clinical trial investigation. Mesenchymal stem cells (*Ma et al., 2023*; *Ahrberg et al., 2018*) and substances such as dexamethasone (*Choi et al., 2020b*) and simvastatin (*Jeong et al., 2018*) administered *via* porous microspheres, though not frequently explored in human trials, present interesting possibilities for evaluation. Previously hindered by adverse effects, innovative approaches, such as dexamethasone and diclofenac, may now offer possible options for clinical studies. Similarly, interventions involving green tea leaves in combination with glycine (*Vieira et al., 2015b*), farrerol (*Wu et al., 2022*), and friedelin (*Jiang et al., 2022*), stand out due to their accessibility. However, despite positive outcomes in preclinical models, treatments like PRP after many human trials have yet to demonstrate satisfactory clinical efficacy (*de Vos et al., 2021*). Furthermore, surgical treatments were underrepresented in the included studies. As the literature shows that patients who fail conservative treatment may have an indication for surgery more preclinical studies investigating the surgical treatments may be warranted (*Maffulli, Sharma & Luscombe, 2004*).

The clinical symptom severity associated with Achilles tendinopathy is not necessarily correlated with their current tendon structure or the extent of tendon damage (*Warden, 2007*). This poses a challenge for translating pre-clinical studies to clinical practice. Pre-clinical studies predominantly focus on the effects of treatments on tendon characteristics,

biochemical effects and histological changes. In clinical practice, factors such as patient expectations, education, concomitant chronic diseases, and coping strategies may play a role in pain severity and quality of life.

## Limitations of the scoping review

Using the ROB tool of SYRCLE, no study was identified that specifically reported how the randomization of the animals were conducted. In systematic reviews that analyse clinical randomized control trials, the specification of the randomization process is an important factor for the strength of the review article (*Sterne et al., 2019*). However, this is not standard practice in animal studies (*Hooijmans et al., 2014*). Some studies did not report crucial data regarding the age or gender of the included animals, which hampers the methodological quality of these studies. Although there is no certainty that the researchers did not consider these points, the absence of reporting these important characteristics diminishes the overall validity of these studies and thus the findings of this review.

Furthermore, a limitation of this review is that a high number of studies were excluded because the treatment was not evaluated on a chemically or mechanically induced AT model but rather on a model where the tendon was either ruptured or surgically damaged. As a consequence, some interesting and innovative treatments and interventions were not included.

Additionally, the literature section, data extraction, and risk of bias evaluation were performed by a single reviewer. This raises the risk of missing relevant articles and misinterpretations during the risk of bias assessment (*Waffenschmidt et al., 2019*). Due to substantial heterogeneity among studies regarding animal and breed differences, initiation methods of the Achilles tendinopathy model, therapeutic application and type, treatment frequency and dosage, follow-up time, and reported outcome measures, no clear conclusion about the best treatment intervention can be reported.

Lastly, a limitation of this scoping review is the absence of preregistration. As mentioned in the method sections, at the time of conducting the review, the authors were not aware of the requirement for preregistration of scoping reviews. Registration of scoping reviews is now recognized as a best practice to enhance transparency and minimize bias. Despite this, we followed the guidelines for scoping reviews, ensuring a systematic and thorough approach of the available literature. Future reviews by our team will integrate preregistration to follow these best practices and further strengthen the transparency of our future studies.

## Implications for future research

This scoping review provides an overview of treatments *in vivo* and *in vitro* studies and may provide thoughts for future preclinical research. Additionally, it enables comparing innovative pre-clinical treatment options with established clinical practices considering AT management. Despite the promising preclinical treatment outcomes, the translation to clinical practice falls behind when comparing the preclinical treatments with common clinical AT guidelines. All the included studies used quadruped animal models for analysis. A justification is needed how quadruped models can be translated to bipedal animal models. Future studies should focus more on standardizing protocols to establish a valid

*in vitro* and *in vivo* animal model and thus strive for less heterogeneity among the studies and in our opinion endeavour to have higher reporting standards to minimize the risk of bias. Less heterogeneity could be achieved by aiming for a universal agreement concerning the histopathological, biochemical, and biomechanical changes. The risk of bias could be minimized using a tool such as the SYRCLE ROB tool to design the animal studies. This may lead to more uniformity of baseline characteristics and a lower risk of bias allowing for comparisons across studies, enhancing their quality, and better understanding of the reported outcomes.

## CONCLUSION

Achilles tendinopathy (AT) is a challenging condition to treat as treatment options lack scientific evidence and have diverse results in clinical practice (*de Vos et al., 2021*). Innovative therapies for Achilles tendinopathy are therefore direly needed. New therapeutic developments predominantly begins with preclinical animal and *in vitro* studies to understand the effects at the molecular level and to evaluate toxicity. This scoping review summarizes the literature regarding *in vitro* and *in vivo* animal studies of AT treatments and evaluates the quality of these studies with the SYRCLE risk-of-bias tool. A total of 98 studies were included which analysed 65 different treatments of which 80% reported promising results regarding the biomechanical, histological, and biochemical outcomes. 33 of the studies reported results in all these domains. Preclinical treatments that improved the biomechanical, histological, and biochemical tendon properties may be interesting for clinical trial investigation. The majority of the included studies had a moderate to high risk of bias according to the SYRCLE risk-of-bias tool. The variety of available data and the quality of the studies display the challenging research domain for pre-clinical AT treatments. These factors may contribute to the lack of translation to the current clinical practice. Preclinical treatments that improved the biomechanical, histological, and biochemical tendon properties may be interesting for understanding the mechanism underlying AT and the testing of innovative therapies.

### Funding
The authors received no funding for this work.

### Competing Interests
The authors declare there are no competing interests.

### Author Contributions
- Nathanael Opoku Agyeman-Prempeh conceived and designed the experiments, performed the experiments, analyzed the data, prepared figures and/or tables, and approved the final draft.
- Huub Maas conceived and designed the experiments, analyzed the data, authored or reviewed drafts of the article, and approved the final draft.

- George L. Burchell conceived and designed the experiments, performed the experiments, authored or reviewed drafts of the article, and approved the final draft.
- Neal L. Millar analyzed the data, authored or reviewed drafts of the article, and approved the final draft.
- Maarten H. Moen conceived and designed the experiments, authored or reviewed drafts of the article, and approved the final draft.
- Theodoor Henri Smit conceived and designed the experiments, analyzed the data, authored or reviewed drafts of the article, and approved the final draft.

## Data Availability

The raw data and codes are available in the uploaded Supplementary Files.

## Supplemental Information

Supplemental information for this article can be found online at http://dx.doi.org/10.7717/peerj.18143#supplemental-information.

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
