# Peer review of "Treatment options for Achilles tendinopathy: a scoping review of preclinical studies"

_PeerJ, doi:10.7717/peerj.18143_

## Round 0.1 · original submission · Major Revisions

Thank you for your considered development of this review article. Whilst the review is well conducted, it was determined that it has limited application within the 'sports medicine and rehabilitation' Section of the journal. However, we have Sections of the journal focused on animal models (rather than applied human research) and this article is suitable there.

Therefore, I am returning the reviewers' comments to you for consideration. If you wish to resubmit, please respond carefully to every comment provided by the reviewers, including what changes you have made in the manuscript in response to the comments. Your response will be sent back to the reviewers for re-consideration.

·

Basic reporting

Original work and topic, good effort, as the matter need attention.

Experimental design

Basic standards maintained

Validity of the findings

Findings are acceptable

Reviewer 2 ·

Basic reporting

The entire manuscript needs to be revised by a fluent English speaker since it contains several flaws in grammar and tenses. Furthermore, check for the correct use of acronyms.

INTRODUCTION
Line 99: what did you mean with "normal activities"?

"AT is multifactorial (genetic, environmental, psychosocial, inflammatory) in nature". Poorly written and confusing sentence. How could you mix up psychosocial and "inflammatory" factors?

Lines 114-117: a concise description of the effects of the factors involved in the pathogenesis of AT (MMP, TIMPS, IL-6, etc) should be reported here for a better understanding of the Results at lines 334-341.

Line 119: these treatments?

Lines 124-127: not pertinent information. Delete.

Line 137, "biomechanical, biochemical, and histological tendon properties". These properties should be clearly explained here or in the Methods section, and not just reported in Tables.

Experimental design

METHODS

The methodology of search is fine. Please just add to point 2.5 that the Table reporting the risks of bias can be found in the Supplementary Material.

RESULTS
Line 260: "these groups"?

Lines 261-262, "The time of analysis and follow-up time of the included studies ranged from 2 hours up to 24 weeks". These information is too vague given the distance between 2 and 24 weeks. Rephrase and re-organize the outcomes evaluation in short-, mid- and long-term outcomes.

Lines 283-297: re-organize these sub-paragraph since it is not easily readable. Put some line space or start new lines.

Line 285: briefly report the "tendon properties".

If only 5% reported negative treatment effects while the 80% reported positive results, what about the remaining 15%?

Lines 311-313: briefly mention the successful interventions, as well the not-successful ones.

Lines 327-330: rephrase them as in lines 311-313.

Sub-paragraph 3.5: this part should be moved at the beginning of the section, after the description of the included studies

Validity of the findings

DISCUSSION
Lines 394-401 should be moved in the "strengths and limitations".

Lines 482-483, "However, only about 4% of the included treatments in this review evaluated exercise therapies for AT". You may also add that the scarce evaluation of exercise therapy should be ascribed to the difficulty of performing any kind of exercise in animal models rather than a low general interest in exercise therapy.

Lines 513-515, "A limitation of this review is that a high number of studies were excluded because the treatment was not evaluated on a chemically or mechanically induced AT model but rather on a model where the tendon was either ruptured or surgically damaged". You may explain why you excluded these kind of animal models, preferably in the Methods section.

Additional comments

When talking about Achilles tendinopathy, please use these references:

https://pubmed.ncbi.nlm.nih.gov/37681821/

https://doi.org/10.14198/jhse.2020.15.Proc4.29

https://doi.org/10.1007/978-3-662-58704-1_32

Reviewer 3 ·

Basic reporting

1. The article is written to a high English language standard.
2. The introduction demonstrates limited rationale for the current study (see general comments).
3. The structure lacks alignment with reporting guidelines (see general comments).
4. The review has limited relevant to humans. I defer to the editor on whether this is relevant for the current journal.
5. Human trials have been meta-analysed extensively, so the rationale for the current study is unclear.
6. Introduction introduces the subject well.

Experimental design

7. As mentioned prior, limited applicable to humans may be an issue.
8. Scoping reviews are a valid method of evidence synthesis, yet it is not clear why this method was chosen and more importantly, why the protocol was not registered a priori.
9. Methods warrant further elaboration (see general comments).
10. Citations are adequate and well paraphrased to my knowledge. I have not run this through an agent such as Turnitin however and defer to the handling editor for internal reports.
11. The review is organised logically, but would benefit from better alignment with reporting guidelines. The discussion in particular would benefit from being more succinct. See general comments for both points.

Validity of the findings

12. It is not clear how the conclusions can be extrapolated to bipedal human models, yet this is done at times. I would recommend avoiding these comparisons.
13. Argument based on goals set out in Introduction is sound for animal models, yet extrapolation to human models is with marked limitation.
14. Conclusions warrant attention (see general comments).

Additional comments

15. The lack of protocol and registration is concerning. Please add details on why this was not done.
16. Thank you for appending the PRISMA-ScR checklist. Some elements, however, are noted as reported, yet not reported in line with checklist recommendations. For example, there is not a structured abstract. Another example, there is no statement on why there is no protocol/registration. The overall structure of the manuscript would benefit from alignment with the reporting guidelines – i.e. subheadings closely aligned with reporting items would benefit flow.
17. Abstract | Please avoid abbreviations.
18. Abstract | Please add rationale for conducting the current review.
19. Abstract | Please refrain from commenting on clinical implications based on animal models. Rather, focusing on what studies could be conducted based on findings from pre-clinical studies is more suitable.
20. Introduction | The rationale for understanding mechanisms beneath current treatments is important, but why animal models? What about mechanistic studies in humans? Mechanisms in animal models do not often translate to humans, and in particular with Achilles tendinopathy, which may in part be borne from the upright bipedal physical activities among humans (as evidenced by the higher prevalence in runners), there is a need to justify how quadruped models can be translated to bipedal models. In fact, it appears all studies in at least Table 1 are quadruped models.
21. Why a scoping review and not systematic review? Currently there is no rationale for this decision.
22. Do we really need to go back to animal models when we have meta-analyses of RCTs that demonstrate clinically meaningful treatments already? For example: https://bjsm.bmj.com/content/55/5/249
23. The search strategy appears highly robust and is reported in sufficient detail within the supplement. I am unsure, however, about the various highlighting within the search strings – what do these different colours mean?
24. How was duplicates handled when uploaded to Rayyan? Were any of the automated features this service offers utilised?
25. Why was EndNote used for screening when Rayyan was used prior?
26. Who performed the screening? This is not clear within the methods. For each role, please note who did what and how. Where there were multiple screeners please note if this was done independently or not.
27. It would be ideal to include information in the main body that supports the conclusions, not just that which examined all outcome measures.
28. Avoid commencing a sentence with a numeral – e.g. line 285.
29. Discussion | No need to introduce abbreviations again.
30. Discussion | The first sentence does not seem supported by evidence. For the most part, there is quite a lot of mechanistic evidence as to why current human treatments work, but even if there was not, how do quadruped animal models answer this question?
31. Discussion | Why would sheep and rabbit models be comparable to humans?
32. Discussion | Overall, this section is quite long and at times is more of a summary of results, rather than discussion of said results.

---

## Round 0.2 · accepted · Accept

Thank you for addressing the previous reviewers' concerns.

Reviewer 2 ·

Basic reporting

No more comments, the Authors improved their manuscript according to my suggestions.

Experimental design

No comment

Validity of the findings

No comment

Additional comments

No comment